# Assessing the Green R&D Investment and Patent Generation in Pakistan towards CO$_2$ Emissions Reduction with a Novel Decomposition Framework

**Muhammad Yousaf Raza \*, Yingchao Chen and Songlin Tang**

School of Economics, Shandong Technology and Business University, Yantai 255000, China; chenyingchao614@163.com (Y.C.); tangsonglin@sdtbu.edu.cn (S.T.)
\* Correspondence: yousaf.raza@ymail.com or yousaf.raza@sdtbu.edu.cn

**Abstract:** Energy plays an imperative role in global economies, such that products and services are generally dependent on energy use. This study leads to the application of environmental policies under green research and development (R&D) investment in Pakistan. Existing research has tried to analyze the effects of R&D investment associated with patent applications using the logarithmic mean Divisia index (LMDI) method called PATENT. The objective of this method is to examine the variations in R&D activities motivated by the reduction of fossil fuel power. The research contributes the following: (1) the R&D reaction is the main factor in raising the number of patent applications, while R&D efficiency needs more enhancements. (2) Reaction and production effects are imperative in raising the number of patent applications during the study period. (3) R&D expenditure presents a significant rise in renewable energy technologies (RETs), by 6.7% yearly, which ultimately impacts the economy, sustainability, and the environment. (4) Energy intensity shows a lowering trend in economic development, which confirms that that share of energy will decline, and that Pakistan will move towards significant contributions. Finally, the results show that raising R&D investments, technology transfer and engendered measures are the authentic approaches to Pakistan's environmental and economic development. Based on the analyzed method, the study recommends that environmental regulation policies' efficiency be incremented by investing and joining them with RETs. Furthermore, the concerned policies linked with the estimated outcomes are provided below.

**Keywords:** green technologies; LMDI; research and development; fossil fuel consumption; climate change; Pakistan

## 1. Introduction

Greenhouse gas (GHG) emissions (including CO$_2$, nitrous oxide, methane, ozone, and water vapor) are leading to global warming and climate change [1–3]. This is due to the fast growth of fossil fuel utilization because of positive results related to emissions [4,5]. These gases influence the natural environment in a negative way. By annual average from 1997 to 2016, the long-term climate risk index (CRI) of Pakistan (CRI 30.50 score, gross domestic product (GDP) loss per unit 0.605%, death rate 523.10, total losses of purchasing power parity USD3816.82 million) among the top ten most affected countries [6]. Recently, R&D (research and development) and modern worldwide technologies have been concentrating on improving the efficiency and economics of carbon dioxide emissions (CO$_2$Es) [7], because the annual changes in the CO$_2$E affect the level of R&D investment for CO$_2$ reduction technologies [8]. In 2016, Pakistan spent 0.29% of its GDP on R&D compared to 0.82% spending by India, 0.94% by Turkey and 1.13%, by Malaysia, while Israel was on top, standing at 4.21% of its GDP. R&D schemes aim to lessen CO$_2$Es, added to about 89% of overall GHG emissions [1,9–12]. R&D and new energy technologies reduce environmental pollution in order to meet the electricity demand, reduce expenses, reduce time, and improve market commercialization [13–16]. Manipulating the most proficient strategy to

avoid worldwide warming and changes in the weather through the examination of the impact of human behavior on modifications in $CO_2Es$ is an important international problem. According to their policies, numerous nations have tried to fight climate change and global warming. However, GHG emission reduction is the target of each country. According to a report by Dinan [17] and Olivier et al. [18], few republics in the European Union (EU) have focused on the maximum decrease of GHGs using innovative technologies. With the Kyoto protocol's promise to reduce emissions and carry out climate mitigation, many countries have applied their policies to promote renewable energy, while the total reduction rate of the EU was 20–30% by 2020 [19,20]. According to a statistical review of world energy [21], the $CO_2Es$ of the world total 33,444 Mt, while Pakistan emits 189.2 Mt with a share of 0.6%, which is very low compared to China (with a share of 27.6%), but a higher percentage than Asian-Pacific countries. Each country's objectives are the environmental assurances that are exceptionally critical. Also, the technologies used for energy efficiency are beneficial in reducing human-related $CO_2E$. Electricity production through the burning of fossil fuels generates 40% of $CO_2Es$. In this way, stored technology has a significant role in generating fossil fuel electricity and improving energy efficiency. For this, Cho and Sohn [1], Kwon et al. [8], and Weixian and Fang [22] analyzed panel data from 1997 to 2007 in order to check the $CO_2Es$ by applying advanced technologies. They found that research, growth and new technologies cause decreasing $CO_2Es$. Many researchers have investigated the influence of new technologies in different sectors (i.e., agriculture, R&D, economics, climate change, firms, etc.); economic growth and technology progression are the primary variables touching $CO_2Es$ [23–28]. The researchers explained that $CO_2Es$ increase due to population and GDP, while the expansion of technology is costly, but it reduces $CO_2Es$. In particular, energy technologies protect the environment, and clean combustion, high-efficiency energy and innovation play an influential part in falling energy-related $CO_2Es$ [29,30].

Therefore, green technologies and patent data have been employed [8,31–34]. $CO_2Es$ and R&D commitment to green technologies have not been considered widely. Many studies in different countries have been written to determine the intensity of $CO_2$ reduction by using Kaya Identity. For this, many countries signed ecological strategies, i.e., the United Nations Framework Convention on Climate Change (UNFCC), the Paris Agreement, and the Kyoto protocol. Therefore, many researchers have focused on new technologies and invested in R&D for the bright-green environment, novelty, and patent applications [35–39]. They showed that climate change positively correlates with advanced technologies, innovations, and patent compliances. Apergis et al. [40], Li and Lin [41], and Wesseh and Lin [2] pointed out that R&D expense has an optimistic effect on the diminution of $CO_2Es$. It also improves industrial energy efficiency and reduces industrial energy consumption [42]. Furthermore, according to Manoli et al. [43] and Obani and Obani [44], green technologies play an essential role in balancing $CO_2Es$ as fossil fuels. For this, the Alternative Energy Development Board (AEDB) was introduced in 2003; it is working for the betterment of green technologies in Pakistan. The main objective is to promote different projects related to renewable energy at the international level, working under the International Solar Energy Society (ISES) and the World Wind Energy Association (WWEA) [45,46]. The European Union also encouraged R&D to lessen $CO_2Es$ [8]. Additionally, through endogenous growth theory, administration involvement with the Kyoto Protocol, UNFCC and the Paris Protocol includes the technical skills related to green innovations [47]. (The endogenous growth theory focuses on positive externalities and a knowledge-based economy, which causes economic development. Initially, an economy's long-run growth rate depends on policy measures. Subsidies on R&D, innovation and growth development in different sectors can lead to positive results. Endogenous growth theory can be employed in the fossil fuel energy division and novel LMDI decomposition analysis [1]. In fact, Tang and Tan [48], Fei et al. [49], and Fernández et al. [50] showed the relationship between endogenous growth theory and production practices, technology innovation, R&D, and economic growth that are friendly in the case of the environment. According to 'EGT', technology and a friendly environment cause economic growth and also improve the capacity of polluting

resources [50]). However, quantitatively, there are not many previous studies about countries' investing plans on R&D for green advancements and their effects. As such, depending on the endogenous growth theory, we have examined the impacts of $CO_2$ variations on green technologies through environmental policies and R&D investment. In the current study, the number of patent applications is considered to be the level of technical progress, as patent data are taken as a proxy for R&D actions, and innovative technologies are established through patents. Thus, the study's main objective is to measure how different factors added to the changes in the country's major driving factors. Based on a new framework, the environmental factors are as follows: (1) The model analyzes the effects of factors' $CO_2$Es, and assesses the policy implications for carbon mitigation. (2) The factors of novel LMDI decomposition are estimated based on Kaya Identity, which is broadly employed to show the impact of economic growth, energy intensity, fuel mix, and R&D on $CO_2$Es. (3) The study objective is to analyze the relationship between population, technology and affluence, which is helpful in order to estimate the impact of economic growth on the environment. The advanced technologies can measure the impact of $CO_2$Es on green R&D investment and the generation of related patents. Finally, the historical trends of each factor will identify the related facts that enhance the development and application of effective $CO_2$E reduction policies to target the energy framework on renewable production.

This research contributes to five aspects. First, the study investigates the effects and aggregate spending on R&D and the use of R&D efficiency in reducing $CO_2$Es during 1993–2017, which has not been checked before in Pakistan. However, the considered factors are complex in the estimation of the green R&D investment and patent generation; we estimated the total association on the availability of data from 1993 to 2017, which is very important for Pakistan's present energy and economic situation. The impacting factors are the economic production, energy intensity, fuel mix, $CO_2$E coefficient, R&D reaction, and R&D efficiency effect. These factors are important due to their need, supply, sustainability, and profitability. Second, the logarithmic mean Divisia index (LMDI) technique was applied for environmental policies associated with GHG emission diminution, which is totally different from the studies, for example, of Jung et al. [10] for South Korea, Lin and Raza [51] for Pakistan, Akbostancı [52] for Turkey, and Huijie [53] for China, who found that economic, energy and social factors are directly or indirectly correlated with each other. Furthermore, the LMDI method is important over the other techniques due to its adaptability, ease of use and theoretical basis [54]. In addition, this method is applied in decomposition variations in $CO_2$Es, investigating the zero-value problem, environmental issues and regional analysis. Before, a patent application had not been applied in Pakistan to estimate the effects of modifications in $CO_2$Es on R&D policies for green technologies. Therefore, this technique motivates us to measure patent data and R&D expenditure related to fossil fuels. We believe that the given model and research is the first study of its nature, to show R&D results and effectiveness as important variables of the $CO_2$E structure. Third, as R&D is directly linked with patent applications, this research is not only supportive of Pakistan's energy study policy but also identifies the $CO_2$ changes on the patent application. For this, R&D investments using ten energy factors—motor gasoline, aviation gasoline, kerosene, diesel oil, and residual fuel oil, gas, and coal—are considered fossil fuel and R&D reaction effects. The R&D efficiency effects are considered as a number of patent applications. Fourth, the calculation period in the current research starts with 1993–2017, which is divided into six planning periods: 1993–1997, 1998–2002, 2003–2007, 2008–2012, 2013–2017, and 1993–2017. Finally, as per the method and fitting line, the predicted outcomes from 2018 to 2040 are estimated, which will help the policymakers, scholars and government to prepare themselves for climate, economy and energy security.

The article's structure is given as follows: Section 2 clarifies the literature related to decomposition analysis. Section 3 contains the new LMDI model. Section 4 discusses the outcomes, and Section 5 sums up the conclusion and policy suggestions.

## 2. Literature Review

Since the economic and industrial revolution, fossil fuel consumption has increased as a primary energy source, which has impacted global climate change, particularly $CO_2$Es, thus affecting various human health diseases. For this, Pakistan has agreed to use renewables and friendly environmental resources. Thus, the decomposition technique is generally applied in order to decompose the $CO_2$Es and estimate various factors' significance.

According to Guan et al. [9], two methods are used for environmental analysis: structural decomposition analysis (SDA) and index decomposition analysis (IDA). Using SDA, numerous pieces of research were applied to reach significant results [20,55–57]. They used the SDA model and explained $CO_2$Es in China's manufacturing sector using three effects, i.e., demand, intensity and technology. Casler and Rose [58] and Liu et al. [59] showed that SDA is based on the input–output (I-O) table, which is proficient in quantifying fundamental "sources" of change in a vast range of variables, including energy consumption, material intensity, economic growth, and $CO_2$Es. The first study of SDA analyzed conventional air pollutants produced by the United States [60], giving a set of pollution coefficients added to I-O tables. This has also been used in energy, emission indicators, and recently in the environment [61–63].

However, IDA is again divided into two indexes, the Laspeyres Index (LI) and Divisia Index (DI). LI is used to solve the residual problem (including extreme residual value) without zero-value problems, and has also been used to study environmental issues since 1995 [64,65]. Both LI and DI techniques have been taken from statistics, mathematics and economics [66–70], using LI to resolve the residual problem. Ang and Zhang [69] showed that if the amount of different factors exceeds three, it can be very complicated. Boyd et al. [54] also suggested DI, which was again extended by several researchers [71–74], who analyzed at the industrial and regional levels, and found that replacing zero values with a small number provides converging outcomes. The DI is divided into 'two' indexes: the arithmetic mean Divisia index (AMDI) and the logarithmic mean Divisia index (LMDI). The LMDI method is applied for decomposition, and is widely used in the analysis of $CO_2$Es; it deals with zero-value problems [1,51,52,75–79]. Thus, the LMDI has been taken as a decomposition technique, and can decompose many issues without residual problems. It can be again divided into two portions of additive and multiplicative decompositions. As per Choi and Ang [80], the additive decomposition decomposes the variation in the amount change, such as real GDP, to various sources associated with this change. Ang and Liu [73] invented multiplicative decomposition, which measures ratio changes. According to Ang and Zhang [69], multiplicative decomposition is helpful in time-series study, while additive decomposition is useful for timewise investigation.

This research is based on an additive decomposition method (ADM) analysis because of time investigation to compare the initial- and final-year period for any country or region, especially for Pakistan. A similar analysis has been completed by [1,81]. The focus of our study is on the environmental issue relating to energy consumption. Many researchers—for example, Kwon et al. [8], Jia et al. [82], Lin and Lei [83], Roman-Collado et al. [84], Donglan et al. [85], and Huijie et al. [53]—focused on the power utilization and GHGs related to environmental problems employing the LMDI decomposition method. These studies gave no direct influence of $CO_2$Es on R&D. After that, for the reduction of $CO_2$Es, green technologies are used by applying the decomposition method. Recently, a few researchers used LMDI techniques to explain the aspects of green technology's progress by decaying the number of patent applications [86,87]. Moreover, the analysis of yearly sharing or changes in patent applications might help us to understand the growing trend of any technological region [88,89]. Researchers have showed that patent applications per unit of R&D investment and expenses are important factors for promoting green technologies. These technological factors were used to suppose that studies could not examine the ecological factors motivating the innovation of green technologies, i.e., $CO_2$ production and energy utilization. As such, the environmental factors are the original cause of R&D investment in cleaner technologies [90,91]. Kwon et al. [8] indicated that a rise in $CO_2$Es leads to a

rise in green patent applications. Fabrizi et al. [92] used green patent applications as green innovations, as many patents have poor technological and economic relevance. Ahmad and Wu [93] investigated the role of green productivity growth, economic globalization and eco-innovation for OECD countries, and found that green productivity growth linearly and non-linearly mitigates environmental degradation. Gao et al. [94] showed that local or neighboring land resource misallocation has an obstructing impact on local green technological development, and even reduces pollution. Moreover, the solar energy technology applications in micro-, small- and medium-enterprises, even at the district level, can reduce pollution [95]. Similarly, Rehman et al. [96] and Shahzad et al. [97] analyzed technological innovation and economic progress, and found significant outputs.

In order to overcome this problem, we are dealing with related patents. In fact, many countries have tried their best to reduce $CO_2$Es and energy utilization through R&D investment. However, research on the relationship between green technology and $CO_2$Es using the LMDI technique that has not been carried out yet in Pakistan. As such, we have introduced an innovative LMDI technique that may be utilized to check variations in R&D actions led by the requirements of the decline of fossil fuels. These are taken as energy associated with $CO_2$Es in expressions of R&D investment and the production of green patents.

## 3. Methodology

### 3.1. Patent Model

The patent model, Equation (5), investigates the different factors used in this model to show Pakistan's technical and environmental efficiency. We have discussed the factors related to $CO_2$Es in Pakistan through this model. All of the the factors have been discussed in LMDI models, but R&D and green patent applications have not been applied before in LMDI models based on Pakistan's specific technologies, as discussed in (Appendix A). We have utilized innovation and technological factors to overcome the $CO_2$Es in Pakistan. Based on the analysis and patent applications, we established the patent model of $CO_2$Es, which can be discussed after knowing the LMDI I, LMDI II and IPAT, as follows.

### 3.2. Association between LMDI I, LMDI II and IPAT

LMDI II and LMDI I provide superior decomposition outcomes without residual terms. The estimates of each effect specified by these techniques will change slightly. Before this, many studies have contrasted these two methods with respect to their strengths and weaknesses. IDA is related to an index number, and both methods are related. Additive LMDI I fails the proportion analysis, while the additive LMDI II fails the aggregate test [98]. IPAT identities are used to expose environmental $CO_2$Es. The objective is to identify the impact of economic growth on atmospheric $CO_2$Es [1,99]. We will find the difference between the three techniques (LMDI I, LMDI II and IPAT). IPAT is our concern, which includes innovation and technological factors. For this, $CO_2$Es from energy consumption can be measured using the Kaya identity, which has been taken to be an excellent method [100,101]. For this, the relationship between the LMDI method and $CO_2$Es, population, and economic policies has been taken from human activities. Energy consumption is higher due to human activities. Therefore, they have large $CO_2$E contributions. Power efficiency plays a significant part in lessening $CO_2$Es; therefore, in order to check and examine the effects of fossil fuel energy consumption, we developed a model as Equation (1):

$$I = P.A.T \tag{1}$$

Equation (2) examines the various factors motivating environmental pollution [101]. IPAT Kaya identity has already been employed in China, Pakistan, OECD countries, and South Korea [1,75,77], and was first introduced by Ehrlich and Holdren [102]. The IPAT model provides the framework to analyze the determinants of the environment based on three shortcomings. Firstly, the model is based on a mathematical formula that cannot test how the different factors affect environmental change [103]. Secondly, the model

shows the relationship between population, affluence, and technology. Thirdly, Ehrlich and Holdren [102] measure the environmental impact. This also tests the effect of economic development on the environment. Additionally, the environmental Kuznets Curve (EKC) presents pollution at first rising and then decreasing as income rises, which has become preserved in standard economic principles [104]. A very different view was set by Ehrlich and Holdren, who showed that IPAT is known as a Kaya identity, which plays a central role in the Intergovernmental Panel on Climate Change (IPCC) estimates of future $CO_2Es$. In these measures, overall $CO_2Es$ are a product of population, GDP per capita, energy use per capita, and $CO_2Es$ per unit of energy consumed. Therefore, most of the scientific community still relies on their famous IPAT equation. Overall, the EKC measures the hypothesis between dependent and independent variables, while the decomposition method analyzes the effects of the various social factors. The related $CO_2E$ factors are given in Table 1.

**Table 1.** Energy-related $CO_2$ emission change variables. The variables are aviation gasoline, motor gasoline, oil, kerosene, diesel, residual fuel oil, LPG, LNG, natural gas, coal, and R&D activities. Here, patents have been taken as green technologies. All of the data and information are collected and analyzed as per the five-year economic plan of Pakistan, which represents the maximum interval on the availability of data. Moreover, the R&D-related data reported as expenditure referred to the interval of 1993–2017. The R&D expenditure is considered as the total % of GDP, as per accessibility.

| Variable | Determinant | Description | Item |
|:---:|:---:|:---:|:---:|
| EI | $\frac{TEC^t}{GDP^t}$ | Energy Intensity Consumption | TEC: Total energy consumption from total GDP |
| SUB | $\frac{FFC^t}{TEC^t}$ | Substitutions (total mixed energy) | FFC: Fossil fuel Consumption |
| CI | $\frac{CO_2{}^t}{FFC^t}$ | Carbon Intensity effect | $CO_2Es$: Carbon dioxide emissions |
| $C_{it}$ | $\frac{R\&D_{it}}{CO_{2\ it-1}}$ | Research &Development Investment | R&D: Innovations and analysis |
| $R_{it}$ | $\frac{PATENT_{it}}{R\&D_{it}}$ | Generation of green patents | PATENT: New technologies for the development of a green environment |
| AC | $\frac{GDP^t}{Pop^t}$ | Activity effect | AC: change in emissions due to per capita change in GDP |

### 3.3. LMDI Approach

The employment of the LMDI method to find out the exact value of $CO_2Es$ based on fossil fuels, energy consumption, GDP and population is presented in Section 3.4.

### 3.4. Patent Model

See below equations.

$$CO_2 = \sum_{i-1}^{9} Population \cdot \frac{GDP}{Population} \cdot \frac{Energy}{GDP} \cdot \frac{CO_2}{Energy} (i = 1....9) \tag{2}$$

The patent model related to R&D is given as

$$PATENT_t = \sum_{i-1}^{11} GDP_{t-1} * \frac{TEC_{t-1}}{GDP_{t-1}} * \frac{FFC_{it-1}}{TEC_{t-1}} * \frac{CO_{2it-1}}{FFC_{it-1}} * \frac{R\&D_{it}}{CO_{it-1}} * \frac{PATENT_{it}}{R\&D_{it}} \tag{3}$$

$$PATENT_t = f(GDP_{t-1} * EI_{t-1} * SUB_{it-1} * CI_{it-1} * C_{it} * R_{it}) \tag{4}$$

$C_{it}$ and $R_{it}$ indicate the number of patent submissions in the fossil fuel energy regions. The calculation of each factor and the total factor is as follows:

$$\left(P_{tot} = P^t - P^0\right)$$

where $\Delta P_{tot}$ is the change in green patents for recovering fossil fuel energy proficiency starting from $P^0$ to the current year $P^1$.

$$\Delta P_{tot} = P(1) - P(0) = \Delta P_{GDP(production)} + \Delta P_{EI(energy\ intensity)} + \Delta P_{Sub(fuel\ mix\ effect)}$$
$$+ \Delta P_{CI(CO_2\ emissions\ coefficient)} + \Delta P_{C(research\ \&\ development\ effcect)} + \Delta P_{R(research\ \&\ development\ efficiency\ effect)} \tag{5}$$

$$\Delta P_{GDP(production)} = \sum \frac{P_i(1) - P_i(0)}{\ln[P_i(1)/P_i(0)]} \ln\left[\frac{GDP_i(1)}{GDP_i(0)}\right] \tag{6}$$

$$\Delta P_{EI(energy\ intensity)} = \sum \frac{P_i(1) - P_i(0)}{\ln[P_i(1)/P_i(0)]} \ln\left[\frac{EI_i(1)}{EI_i(0)}\right] \tag{7}$$

$$\Delta P_{Sub(fuel\ mix\ effect)} = \sum \frac{P_i(1) - P_i(0)}{\ln[P_i(1)/P_i(0)]} \ln\left[\frac{Sub_i(1)}{Sub_i(0)}\right] \tag{8}$$

$$\Delta P_{CI(CO_2\ emission\ coefficient)} = \sum \frac{P_i(1) - P_i(0)}{\ln[P_i(1)/P_i(0)]} \ln\left[\frac{CI_i(1)}{CI_i(0)}\right] \tag{9}$$

$$\Delta P_{C(research\ \&\ development\ effect)} = \sum \frac{P_i(1) - P_i(0)}{\ln[P_i(1)/P_i(0)]} \ln\left[\frac{C_i(1)}{C_i(0)}\right] \tag{10}$$

$$\Delta P_{R(research\ \&\ development\ efficiency\ effect)} = \sum \frac{P_i(1) - P_i(0)}{\ln[P_i(1)/P_i(0)]} \ln\left[\frac{R_i(1)}{R_i(0)}\right] \tag{11}$$

where '$i$' is the fuel, '$t$' is the time, R&D is the research and development, $C_{it}$ is the R&D reaction, $R_{it}$ is the R&D efficiency, $PATENT_t$ is the green competence of fossil fuel power, $GDP_{t-1}$ is the financial and economic development, $\Delta P$ represents the change in patents with regard to R&D, $EI_{t-1}$ is the energy intensity, and $SUB_{it-1}$ is the substitution. It is the ratio between FFC and TEC; $CI_{it-1}$ is the carbon intensity with periods. The model examines the variations in R&D activities motivated by the reduction of fossil fuel power regarding research and development investment and the creation of green patents. According to Ehrlich and Holdren [102], IPAT provides a simple theoretical framework to analyze the given factors of the environment. $CO_2$ emissions, GDP, TEC, FFC, R&D, and Patents have been taken as variables to find out both emissions and environmental effects.

The aim is to interpret the outcomes regarding R&D saving and environmental policy. A new LMDI decomposition framework is given in (Figure 1) based on endogenous growth theory. It depends on the country's environmental policies for R&D and technology expansion investments. Unlike the other LMDI frameworks, we supposed both factors for new LMDI decomposition in which R&D is directly linked to patent applications for green technologies. As such, according to Kwon et al. [8], R&D investment variations arise due to changes in $CO_2$Es.

In Equation (3), we have not included the population variable due to a weak relationship with a patent. Due to environmental change, the time interval, and R&D investment, we used the LMDI formula, including GHGs financing and $CO_2$Es. It can be implemented using micro-data that disclose how the product or machinery used with green patents has contributed to the reduction of $CO_2$Es in the companies that bought them. Kwon et al. [8] has also pointed out that such research investments for $CO_2$ reduction technologies are based on the previous year of $CO_2$Es. Therefore, we considered a one-year time lag. Few pieces of research have a time interval of 1–2 years for R&D and patent applications [105,106]. Many researchers, such as Hall and Hausman [107], Nakamura [108], and Brunnermeier and Cohen [109], found that within one year of R&D investment, the patent applications are applied. They also explained that there is no time difference between R&D investments and ecological patent uses. Thus, we did not use the time gap between $CO_2$Es as power utilization. As such, R&D and patent applications happen at a similar time.

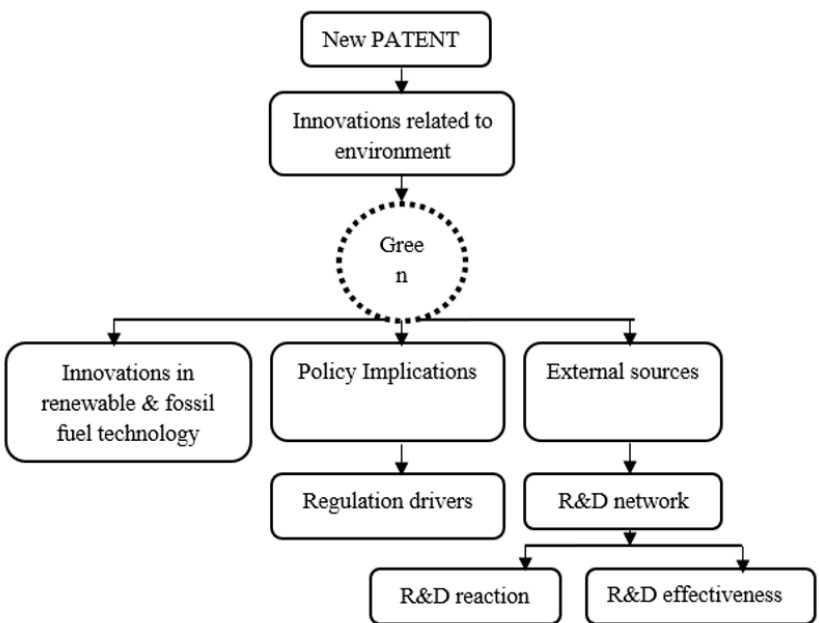

**Figure 1.** The framework of the new patent decomposition method for Pakistan.

*3.5. New Patent Framework*

New patent framework under the new PATENT application is shown in Figure 1.

*3.6. Statistics of the Variables*

Data Source

The data are based on the observations for the number of patent grants by technology applications covering the period 1993–2017. All of the data were collected from the WIPO statistics database, the Statistical Review of World Energy and Pakistan Economic survey. Depending upon the quantity of the different types of fossil fuel consumption and their $CO_2$Es factors from the Pakistan charts and reports, we calculate the $CO_2$E for the active patent grants. The statistical depiction of each variable in the model is shown in Table 2.

**Table 2.** Data sources and variables.

| Variables | Unit | Sources | Mean | Standard Deviation |
|---|---|---|---|---|
| $CO_2$E | Mt | Statistical Review of World Energy | 133.962 | 668.067 |
| FFC | Mt | Statistical Review of World Energy, Pakistan Economic survey | 30,362.8 | 76,490,474.48 |
| TEC | Mt | Pakistan Economic survey; Pakistan Energy Yearbook | 30,604.346 | 78,038,539.91 |
| GDP | Rs. Billion | World Bank; Pakistan Economic survey | 16,914.921 | 179,577,674.1 |
| C | % of GDP | World Bank | 0.0024412 | 0.002012 |
| R | No. of Patent applications | WIPO database | 18.2381 | 213.613 |

Note: Fossil fuel consumption, FFC; total energy consumption, TEC; Rs., rupees.

The green patent applications from Pakistan have been taken in this research, because Pakistan has already signed the COP-21 in Paris for the reduction of emissions. According to INDC, $CO_2$E roots are found in Pakistan [110]. The Kyoto protocol has been used in developed countries, such as the USA, Canada and China, to reduce $CO_2$Es. The European nations have already stepped up the initiative to decrease GHGs emanations using advanced expertise [1,111]. Germany, France, Italy, and the United Kingdom also used patents for OECD due to their R&D investment, green technologies, and higher patent production than the other European countries. Therefore, we have limited our study data to Pakistani economic development organizations, in order to investigate the data of $CO_2$Es from fossil fuel energy on the fluctuations of R&D effects.

The LMDI decomposition analysis in energy-related $CO_2E$ change variables is shown in Table 1, and the data sources are given in Table 2. In this paper, we have used the data of patent grants for technology based on the number of patent applications only in Pakistan. This study is imperative in order to recognize the effects of R&D investment strategy on the expansion of green technologies. Earlier studies have used LMDI techniques for green technologies [1,87]. According to Lanzi et al. [112], the International Patent Classification (IPC) codes, such as the WIPO origin code (given in Appendix A), are linked to green technologies obtaining better fossil fuel energy efficiency for power. These advances depend on the variations in fossil fuel consumption and $CO_2Es$, even in Pakistan. Using Equation (11), we can find Pakistan's fossil fuel consumption and green patent technologies. The given data of all of the fossil fuel variables, R&D and patent applications are analyzed. The European countries measured the impact of each factor using the Kyoto Protocol, with an interval of eight years. Therefore, by the national economic plan of Pakistan, we have analyzed data with a range of five years plans from 1993–97, 1998–2002, 2003–2007, 2008–2012, 2013–2017, and 1993–2017.

## 4. Results and Discussion

The results are presented in three sections. Firstly, we used the patent approach based on R&D, which concerns green technologies' innovation and technological factors. Secondly, we implemented the LMDI technique by using patent applications. Finally, the base-year patent technologies and 2040 forecast show attractive results for the green environment.

### 4.1. Time Grid of Patent Applications

The patent application time graph for green technologies from 1993 to 2017 is shown in Figure 2. The number of patent applications in Pakistan improved quickly from 2009 to 2013, by 7%. It declined after 2013–2015 by a ratio of 1.25, and then again increased until 2017. These patent applications show that a number of new technologies are developing each year. Technological applications grew at a rate of 5.8% from 1993 to 2017. Thus, the national patent trend is continuously motivated from 2012–2017. Patent technologies were considered the least during the period of 1993–2017. This time lag has given effective and significant results for the future period. Moreover, computer technology, turbines, chemical utilization, engineering, transport, and biotechnology positively impact $CO_2Es$. If the trend of patent investment in electrical machinery, apparatus, energy, and computer technology increases, emissions can be controlled.

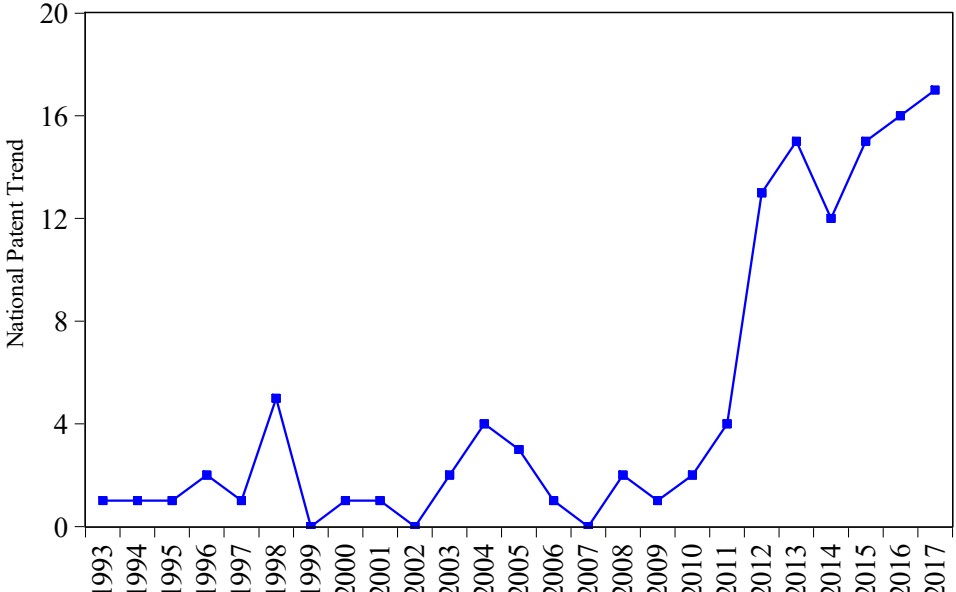

**Figure 2.** The patent trend for green advancement from 1993 to 2017.

*4.2. LMDI Analysis*

Table 3 gives the additive LMDI decay analysis outcomes of green patent applications in Pakistan. The outcomes of all of the energetic factors on green patent applications from the period of 1993–2017 are mixed. The change in the number of patent applications for each unit change in R&D (see Table 3) negatively affects all years. Every aspect communicates the variation in different patent applications. The results show that R&D efficiency effects have a negative impact (9.08%), while the R&D reaction effect has a positive impact (27.81%) on the patent applications. For instance, the developing nations, including Germany, Italy, and UK, found that R&D reaction growth is the major factor in raising the number of patent applications with improvements in R&D efficiency; however, this was not the case in France [1]. In general, the patent applications improved by 89.32% in the interval period of 2013–2017, and by 19.37% from 2008 to 2012 because of production outcomes. While the energy intensity (EI) effect and the fuel mix effect led to decreased patent production, the $CO_2E$ coefficient and R&D reaction effect added to the boost in patent production in Pakistan. The R&D efficiency effects directly offset the R&D reaction effect during 1993–2017. From 2013 to 2017, the production effect and $CO_2E$ coefficient result contributed to increased patent production. The EI and fuel mixed effect added to a decline in patent production. R&D effectiveness consequences lead to a rise in patent production in Pakistan, which does not show strong consistency in major contributions, but the overall results of R&D efficiency are useful in green patents. These outcomes are consistent with the studies of Ahmad and Wu [93], and Raza and Lin [113]. The R&D reaction and fuel mix improved the patent production from 2008 to 2012 and decreased the patent production from 2013 to 2017, but the overall effects of both variables from 1993 to 2017 increase the patent production. However, this had comparatively little influence on variations in current patent production. These outcomes illustrate that the impact of a few indicators on patent applications, i.e., R&D efficiency during 1993–2017 is 49.04%; the R&D reaction 13%; and the $CO_2Es$ coefficient, energy mix effect and energy intensity effects altered suddenly during the studied periods.

**Table 3.** Individual factors' effects on green patent applications in Pakistan, 1993–2017.

| Year | $\Delta P_{GDP}$ | $\Delta P_{EI}$ | $\Delta P_{Sub}$ | $\Delta P_{CI}$ | $\Delta P_C$ | $\Delta P_R$ | $P_{tot}$ |
|---|---|---|---|---|---|---|---|
| 1993–1997 | 0 | 0 | 0 | 0 | 0 | 0 | 1 |
| 1998–2002 | 0 | 0 | 0 | 0 | 0 | 0 | 0 |
| 2003–2007 | 0 | 0 | 0 | 0 | 0 | 0 | 0 |
| 2008–2012 | 160.03082 | −127.47303 | −0.58796 | −32.12633 | −317.69252 | 1416.4214 | 6.5 |
| 2013–2017 | 737.79012 | −7796.4386 | −3.20496 | 7806.6739 | −8820.6716 | 199.6246 | 1.133 |
| 1993–2017 | 825.9516 | −1477.2918 | −0.55370 | 1195.57503 | −1142.2694 | 2198.2692 | 17 |

4.2.1. Base-Year Analysis

Base-year trend analysis (Figure 3) shows the factor's information. Pakistan should strongly ensure $CO_2E$ reduction in the proposed INDC and COP-21 agreements. Pakistan is currently working harder to develop green technologies to reduce $CO_2Es$ at the initial level, with the cooperation of the developing world. For example, CPEC can play an important role in indigenous energy production—i.e., coal, gas, renewables, RETs, and the Paris agreement—ultimately reducing oil imports and enhancing climate quality [114]. In Figure 3, all of the factors significantly increased patent applications by adding 6.5 technical applications in 2008–2012, while 1.133 applications were added in 2013–2017 (Table 3). The overall number of patents had a great change of 17 patent applications from 1993 to 2017. The rise in the number of patent applications, especially green patents, results in significant output.

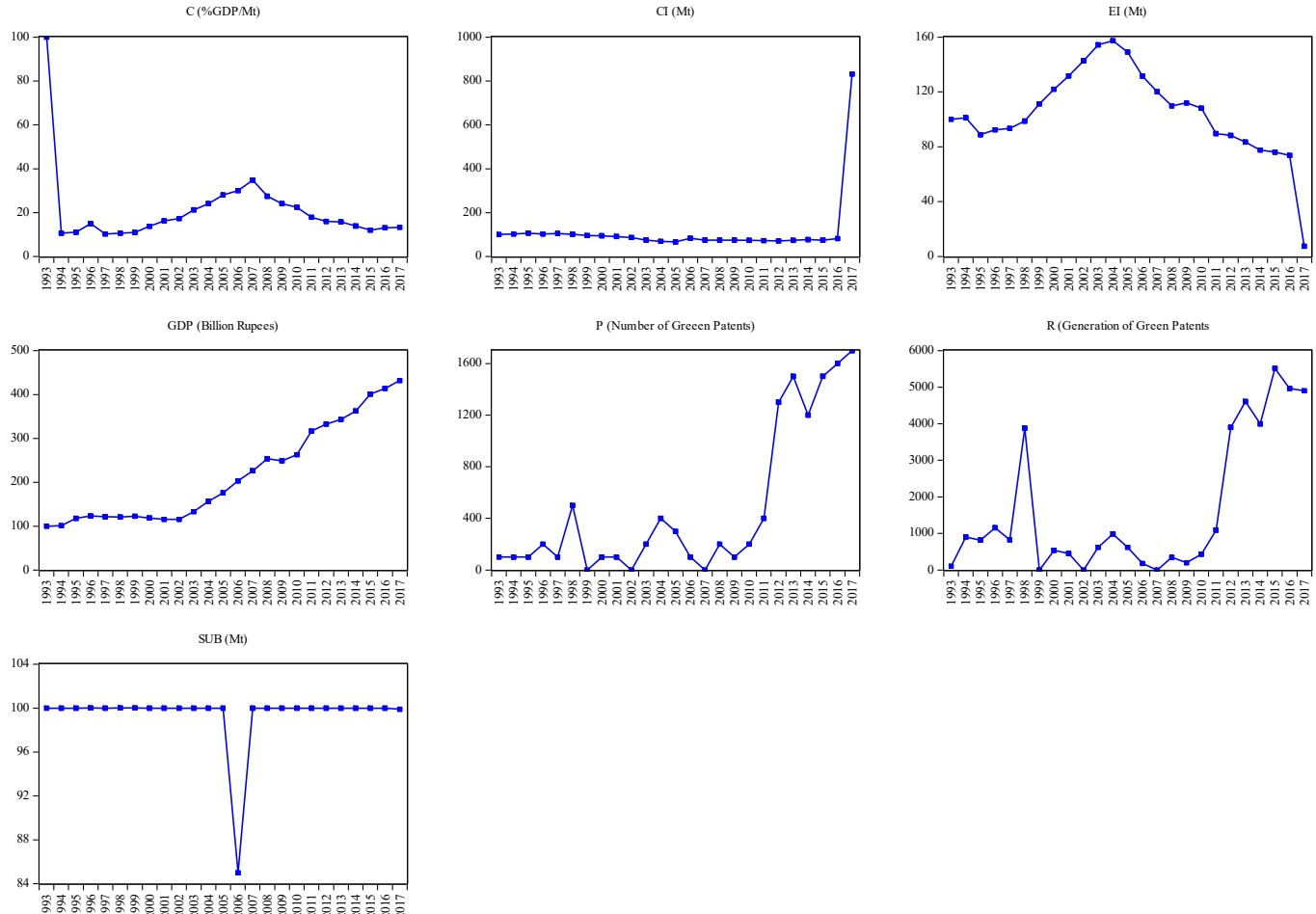

**Figure 3.** Base-year trend 1993–2017.

4.2.2. Forecast of the Individual Increase in Variables

Figure 4 shows the forecast change in factors from 1993 to 2040. Green patents are increasing significantly in the country's gross domestic product. Specifically, the enhancement in patent applications is primarily credited to the R&D reaction. In comparison, the 'EI' and R&D efficiency effects (EE) abridged patent applications. On the other hand, the energy mix (EM) and $CO_2E$ coefficients did not play the most important role in the growth of green technologies from 1993 to 2017. The forecast results (Figure 4) show many patent contributions to Pakistan's technology and climate development according to existing values. The R&D reaction and production effects are taken. Important factors are adding to the rise in the number of patent applications. In particular, the upsurge in patent applications is largely accredited to the R&D reaction effect. The increase in the number of patent applications in Pakistan was little, though Pakistan produced 120 patent grants for technology applications during 1993–2017. However, the 'EI' and the R&D efficiency lessened the number of patent applications [113]. According to the results (Table 3), the additional factors, 'EI', the energy mix, and the R&D reaction effects did not play a key role in the progress of green technologies.

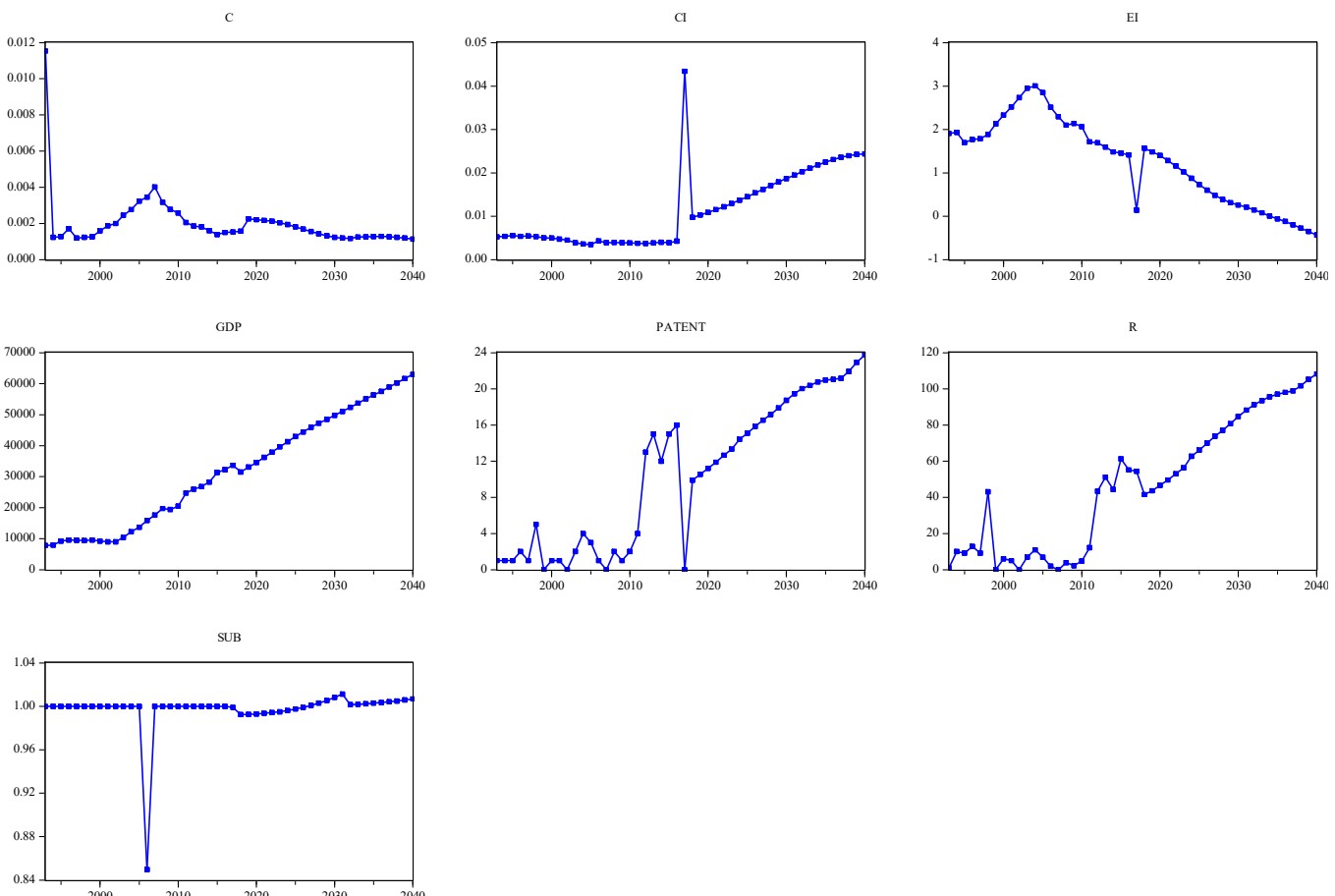

**Figure 4.** Forecast values of all factors from 1993 to 2040.

## 5. Conclusions and Policy Recommendations

### 5.1. Conclusions

Due to energy use, $CO_2E$ has become a severe issue in Pakistan. Pakistan is making important efforts to create patents for green technical advancements which help to improve fossil fuels' energy efficiency. With the production of green technologies, energy efficiency can be improved [8]. The identified variables of novel LMDI decomposition are based on Kaya identity, which is broadly applied to show the influence of economic growth on $CO_2Es$. These factors have given interesting outcomes concerning the variables impacting technological growth.

The empirical results show that (a) R&D response development is the major factor in growing patent applications, and R&D efficiency was required to make enhancements. (b) The patents taken from R&D reaction effects and production effects are taken as important variables adding to the rise in the number of patent applications. As per environmental policy, Pakistan has concentrated on investment in RET development under the China–Pakistan Economic Corridor (CPEC) to reduce imported fuel and pollution [51]. (c) The rise in expenses in R&D shows a significant rise in RETs by 6.67% annually, which shows that R&D would rise, and ultimately impact economic development and the environment. (d) The product impact shows a rise in green technologies; the energy intensity impact shows a rising trend in economic growth. Lowering the energy intensity effect shows that the share of overall energy in the country's economy decreased, showing that this will further Pakistan's progress towards significantly lessening its total energy consumption. Consequently, we have applied a new LMDI technique that can inspect transformations in R&D actions resulting from $CO_2Es$ in provisions of R&D investment and the creation of green patents.

### 5.2. Model Implementation and Policy Suggestions

A new LMDI decomposition technique was used by Kaya identity to signify the impact of economic growth on $CO_2$Es. This study critically examines the factor's implications on patent applications. The decomposition investigation of green advancements in the energy divisions of Pakistan gives attractive outcomes concerning the factors influencing technological progress. The consequences show that the R&D reaction enhancement is the crucial driver of rising patent applications, and R&D effectiveness needed more improvement. Unlike other countries—i.e., France, Italy, Europe and Germany—the R&D reaction negatively affects the expansion of green technologies [1]. In Pakistan, innovation is at its early stage and, more particularly, in small and medium enterprises. This research indicates that Patent Applications show positive and significant impacts on climate change, $CO_2$Es SME (small and medium enterprise) growth, and economic development [115].

Specifically, the patents from the R&D reaction's impact reduced after 2012. According to the environmental policy of Pakistan, Pak-INDC expresses the major challenges faced by the country related to climate change and natural disasters. These actions represent Pakistan's contribution to the global efforts to stabilize the concentration of GHG emissions into the atmosphere. According to CRI-2018, Pakistan is at the seventh position, and lost its GDP per unit by 0.61% [6]. R&D expenses improved by approximately 13.23%, and the number of patent applications connected to renewable energy technology (RET) increased by about 6.67% each year, and by 30.76% from 2011–2016. The R&D expenditure and patent technologies have been taken in this study to reduce environmental emission reduction policies. These technical supports and results affect environmental efficiency. The GDP production in Pakistan can boost green technologies, and the 'EI' effect can reduce green technologies. The decrease in green technologies from the 'EI' effect means that the share of total power utilization in GDP has reduced. The factors of the R&D mix and $CO_2$Es coefficients have a smaller impact on the variation of patent applications.

The outcomes not only efficiently complement the current study; it is imperative for policymakers in Pakistan to pay attention to the findings. Firstly, R&D and green technologies should be highlighted. Secondly, we have proposed some measures for Pakistan based on our results. According to the analyzed results, the patents from R&D reaction effects are quick. Pakistan must invest in R&D (fossil fuel energy) resources in order to produce green technologies. On the other hand, the government should enhance low emission technology by employing modern instruments, i.e., by subsidizing energy companies and associated research institutions to engage in the R&D of low-emission technologies. The government should permit the introduction of different modern clean-energy technologies. Moreover, the government should also progressively reduce renewable energy costs in order to align with the national and international market. This will encourage industries to find ways to develop production efficiency, and reduce energy consumption and $CO_2$Es. Thirdly, the government should produce economic, innovative and green patent-related environmental technologies. Through these technical methods, we can advance the efficiency of patent production based on R&D spending. These patents can also be useful in the reduction of $CO_2$Es. The given method can tell us about the most affecting factors, and boosting interest in R&D can be effective for technology and environmental policy. Fourthly, the endogenous growth theory can be employed in the new decomposition method and fossil fuel energy sectors. However, our study helps the government and fossil fuel-producing companies to lessen $CO_2$Es and generate green technologies. Our study is related only to Pakistan. However, many researchers considered OECD countries regarding the easy access to data. The countries (i.e., China, France, Germany, Italy, and Canada) have already analyzed green patents using the Kyoto protocol for $CO_2$ reduction. Therefore, we have extended this individually in Pakistan by the technical total patent applications to reduce emissions. According to (Figure 3), all of the factors show significant results.

Finally, despite the contributions provided by the present research, certain limitations would guarantee discussion. Firstly, the exact distributions of natural resources due to the vast area of the country are suitable for economic development. Secondly, industries

and technologies should be advanced in both urban and rural areas, because rural areas are populated, and most of the country's income comes from the rural areas (agricultural industry). Therefore, industrial $CO_2$Es can be controlled in different regions at the lower and average levels in order to stay green. Considering these affecting factors can help us to reduce $CO_2$Es in Pakistan. Finally, future studies should be based on capital substitution in the form of renewable energy and RETs; for example, CPEC energy-related projects worth $33.8 billion would change the future scenario of Pakistan. This will play an essential role in environmental sustainability and RETs in the country's economy, reducing $CO_2$Es.

**Author Contributions:** M.Y.R.: Conceptualization, methodology, software, validation, formal analysis, investigation, resources, data curation, writing—original draft preparation, review and editing, visualization, supervision; Y.C.: Validation, formal analysis, investigation, review and editing; S.T.: Validation, formal analysis, investigation, review, editing, supervision. All authors have read and agreed to the published version of the manuscript.

**Funding:** This paper was supported by the Doctors Start Funding Project (No. BS202137) of Shandong Technology and Business University, Yantai, Shandong, 255000, China, the National Social Science Fund of China (Grant No. 21BJY113).

**Institutional Review Board Statement:** Not applicable.

**Informed Consent Statement:** Not applicable.

**Data Availability Statement:** Data can be provided on request.

**Conflicts of Interest:** The authors declare no conflict of interest.

## Appendix A

**Table A1.** Patent grants by technology for the reduction of energy utilization.

| | Description | WIPO Origin Code |
|---|---|---|
| Electrical machinery, apparatus, energy | Energy efficiency, reduce the overall environmental impact and cost to operate equipment. Advancements in electric motor design and modern automation equipment can be extremely energy efficient. | PK-1 |
| Telecommunications | Benefits of data aggregation on energy consumption networks | PK-3 |
| Chemical engineering | Process integration, heat integration, energy integration and pinch technology | PK-23 |
| Environmental technology | Benefits of clean energy. Reduced air pollution and greenhouse gas emissions | PK-24 |
| Digital communication | Stunning advances in data, analytics and connectivity are enabling a range of new digital applications such as smart appliances, shared mobility, and 3D printing | PK-4 |
| Computer technology | Green computing, recyclable and implementing energy efficient technologies | PK-6 |
| IT methods for management | Energy efficiency technologies and energy management practices | PK-7 |
| Measurement | Cost-efficient approach | PK-10 |
| Control | Energy use in different places | PK-12 |
| Medical technology | Historically, equipment designers have paid little attention to energy consumption in electrical devices due to the low cost of energy in the developed world | PK-13 |
| Organic fine chemistry | Techniques for the Manufacture of Organic Fine Chemicals | PK-14 |

**Table A1.** *Cont.*

| | Description | WIPO Origin Code |
|---|---|---|
| Biotechnology | Contribute to the fossil fuel industry by assisting the production of fossil fuels, upgrading fuels, bioremediation of water, soil, and air. | PK-15 |
| Pharmaceuticals | Manufacturing facilities, and other buildings to reduce energy consumption while maintaining or enhancing productivity | PK-16 |
| Food chemistry | Direct energy use for crop management and indirect energy for fertilizers, pesticides and machinery production | PK-18 |
| Basic materials chemistry | Control and fundamental understanding of the chemistry are of paramount importance for the design of new energy-related materials | PK-19 |
| Materials, metallurgy | The reduction of GHG emissions from manufacturing, the environmental impact of the whole powder metallurgy production | PK-20 |
| Surface technology, coating | Coating and plating services provide overcome corrosion, release, wear and friction challenges for oil and gas, mining, food and drink equipment | PK-21 |
| Handling | Governments, businesses and individuals all play a role | PK-25 |
| Machine tools | Advanced machine tool technology can be used as a highly effective energy saving tactic | PK26 |
| Engines, pumps, turbines | Gas-turbine plants; air intakes for jet-propulsion plants; a controlling fuel supply in air-breathing jet propulsion plants | PK-27 |
| Textile and paper machines | Industrial consumption and modern machines | PK-28 |
| Other special machines | Fewer consumption machines | PK-29 |
| Transport | Domestic Electricity Saving Measures | PK-32 |
| Furniture, games | Improving efficiencies and identifying areas of improvement | PK-33 |
| Civil engineering | High-efficiency equipment and automatic controls to minimize energy | PK-35 |

Note: PK, Pakistan.

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
