# Peer review of "Assessing the Green R&D Investment and Patent Generation in Pakistan towards CO2 Emissions Reduction with a Novel Decomposition Framework"

_sustainability, doi:10.3390/su14116435_

Round 1

Reviewer 1 Report

Thank you for the opportunity to review the article. His reading was very interesting. I leave it to the authors' discretion to transfer the results of empirical research from the conclusion to the main part. The conclusion usually refers to the main research findings and collates them with other research. However, tables or other figures look better in the main part of the article. Of course, this is only a suggestion for change, and I leave the decision to the authors.

Reviewer 2 Report

The article is an interesting attempt to link the level of innovation and its potential to reduce GHG emissions. 
Comments and Suggestions for Authors:
1. The article lacks a clear statement of the research objective.
2. Forecast to 2040. Forecasting any values, especially numbers of patents, over such a period is inappropriate. 
3. It was difficult for me to find information whether the calculation uses expenditures on R&D. It seems to me that it does not. It is only in line 375 that information on research outlays appears. In such a case one cannot speak of investment in research but only of the number of patents. Of course the number of patents is dependent on expenditures, but these are two different quantities.  We then have a completely different title and aim of the article. I would like to ask the authors to explain in detail whether research expenditures were taken into account in the construction of models. This is crucial for the article. 
4. A wider comment: the problem is the use of filed patents in research. The question is how many of them were actually applied? 

Author Response

Response are attached.

Reviewer 3 Report

The writing of the manuscript is confusing and mixes too many topics without guiding the reader on the main issue. There are 107 articles referenced!
The introduction should be completely reworded.
The literature review is poorly numbered (1. Literature review) and includes too many articles that deviate from the main topic that the work wants to address.
The question that the authors want to answer is very interesting. However, they try to answer it with data aggregated at the sector level and by assuming certain relationships as true (see equation 3).

How could they show that these relationships are valid? The answer is simple: only with microdata that reveals how the use of products or machinery made with the green patents have contributed to the reduction of emissions in the companies that bought them.

Author Response

Response are attached.

Reviewer 4 Report

I read the manuscript " Assessing the Green R&D Investment and Patent Generation in Pakistan Towards CO2 Emissions Reduction With a Novel Decomposition Framework" with care and attention and have the following observations:

1) The manuscript deals with an important issue; however, certain concerns need authors’ attention before the paper is accepted for publication by Sustainability.

2) Abstract needs modifications: Please add a more relevant background sentence in the beginning of the Abstract followed by a sentence stating the problem statement of this research. The contributions, either theoretical or empirical, made by this study should be reported in one or two sentences in the Abstract. At the end of Abstract, a take-home message should be provided in the form of a practical policy implication.

3) The motivation of research is clear; however, the contributions paragraph in the end of Introduction section should be improved further.

4) Following studies presenting nexus of technological innovation, R&D and green investments with carbon emissions around the world would help strengthen the background and literature review of this research.

Ahmad, M., Wu, Y., 2022. Combined role of green productivity growth, economic globalization, and eco-innovation in achieving ecological sustainability for OECD economies. J. Environ. Manage. 302, 113980. https://doi.org/10.1016/j.jenvman.2021.113980

Ahmad, M., Zhao, Z., 2018. Causal linkages between energy investment and economic growth : a panel data modelling analysis of China growth : a panel data modelling analysis of China. Energy Sources, Part B Econ. Planning, Policy 00, 1–12. https://doi.org/10.1080/15567249.2018.1495278

Anser, M.K., Ahmad, M., Khan, M.A., Nassani, A.A., Askar, S.E., Zaman, K., Abro, M.M.Q., Kabbani, A., 2021. Progress in nuclear energy with carbon pricing to achieve environmental sustainability agenda: on the edge of one’s seat. Environ. Sci. Pollut. Res. 28, 34328–34343. https://doi.org/10.1007/s11356-021-12966-y

Atchike, D.W., Irfan, M., Ahmad, M., Rehman, M.A., 2022. Waste-to-Renewable Energy Transition: Biogas Generation for Sustainable Development. Front. Environ. Sci. 10, 1–9. https://doi.org/10.3389/fenvs.2022.840588

Fareed, Z., Rehman, M.A., Adebayo, T.S., Wang, Y., Ahmad, M., Shahzad, F., 2022. Financial inclusion and the environmental deterioration in Eurozone: The moderating role of innovation activity. Technol. Soc. 69, 101961. https://doi.org/10.1016/j.techsoc.2022.101961

Gao, X., Wang, S., Ahmad, F., Chandio, A.A., Ahmad, M., Xue, D., 2021. The nexus between misallocation of land resources and green technological innovation: a novel investigation of Chinese cities. Clean Technol. Environ. Policy. https://doi.org/10.1007/s10098-021-02107-x

Iqbal, N., Abbasi, K.R., Shinwari, R., Guangcai, W., Ahmad, M., Tang, K., 2021. Does exports diversification and environmental innovation achieve carbon neutrality target of OECD economies? J. Environ. Manage. 291, 112648. https://doi.org/10.1016/j.jenvman.2021.112648

Qamar, S., Ahmad, M., Oryani, B., Zhang, Q., 2022. Solar energy technology adoption and diffusion by micro , small , and medium enterprises : sustainable energy for climate change mitigation. Environ. Sci. Pollut. Res. https://doi.org/10.1007/s11356-022-19406-5

Rehman, A., Ma, H., Ahmad, M., Ozturk, I., Işık, C., 2021. Estimating the connection of information technology, foreign direct investment, trade, renewable energy and economic progress in Pakistan: evidence from ARDL approach and cointegrating regression analysis. Environ. Sci. Pollut. Res. https://doi.org/10.1007/s11356-021-14303-9

Shahzad, F., Ahmad, M., Fareed, Z., Wang, Z., 2022. Innovation decisions through firm life cycle: A new evidence from emerging markets. Int. Rev. Econ. Financ. 78, 51–67. https://doi.org/10.1016/j.iref.2021.11.009

5) Why did the authors choose to employ STIRPAT framework and why not other reputed frameworks such as EKC?

6) What are the grounds for applying LMDI approach in the current case study?

7) What is 1.1.1 in the beginning of Equation 4? It seems there are several typo errors needing authors’ attention here.

8) The figure 1 should be properly framed. Current figure is not understandable.

9) The conclusions section should not have any equations. This section should single out the main conclusion points in the first paragraph, followed by the policy implications, limitations of this study and directions for the future studies.

10) The discussions are currently weak. Please discuss your main findings with in-depth critical evaluation of your results in the light of past studies.

11) The language quality should be improved further to meet the high-quality journal readership such as Sustainability.

12) Once the above concerns are addressed, the manuscript could be reconsidered for publication by Sustainability.

Author Response

Response are attached

Round 2

Reviewer 4 Report

Since the authors have thoroughly and sufficiently addressed my concerns, I am happy to recommend the paper for publication in its present format.